# Albic Podzols of Boreal Pine Forests of Russia: Soil Organic Matter, Physicochemical and Microbiological Properties across Pyrogenic History

Alexey A. Dymov [1,2,*], Irina D. Grodnitskaya [3,4], Evgenia V. Yakovleva [1], Yuri A. Dubrovskiy [1], Ivan N. Kutyavin [1], Viktor V. Startsev [1], Evgeni Yu. Milanovsky [5] and Anatoly S. Prokushkin [3,6]

[1]  Institute of Biology of Komi Scientific Center Ural Division of Russian Academy of Science, Syktyvkar 167982, Russia
[2]  Department of Physics and Soil Reclamation, Faculty of Soil Science, Lomonosov Moscow State University, Moscow 119991, Russia
[3]  V.N. Sukachev Institute of Forest SB RAS, Krasnoyarsk 660036, Russia
[4]  Institute of Fundamental Biology and Biotechnology, Siberian Federal University, Krasnoyarsk 660041, Russia
[5]  Soil Physicochemistry Laboratory, Institute of Physicochemical and Biological Problems in Soil Science RAS, Moscow 142290, Russia
[6]  School of Ecology and Geography, Siberian Federal University, Krasnoyarsk 660041, Russia
*  Correspondence: aadymov@gmail.com; Tel.: +7-82-1224-5115

**Abstract:** Albic podzols under pine forests are more prone to fires on the planet. The influence of fire extends to all soil components, including chemical properties, microbiological characteristics, and the composition and structure of soil organic matter, which persists for a long time. Here, we present the results of a study of the morphological, physicochemical, and microbiological properties and features of soil organic matter (SOM) in the albic podzols of pine forests (*Pinus sylvestris* L.) not exposed to fires for a long time (from 45 to 131 years). The study areas are characterized by different numbers of old fires (from four to five) that occurred over the previous several centuries in various territories of the Russian Federation such as Central Siberia (CS) and the European North (EN). In general, the albic podzols developing in CS and EN are characterized by similar morphological and physicochemical properties, with high acidity and poor mineral horizons. In terms of the lower vegetation layer and stand parameters, forest communities at the CS sites have a lower density and species diversity than those in EN. The ground cover is almost completely restored 45 years after the surface fire. The upper mineral horizon of albic podzols in EN contains higher PAHs in comparison with similar horizons of the CS sites. In the soil of EN pine forests, the MB content in the mineral horizons is, on average, three times higher than those in CS. Differences were also found in the qualitative composition of the studied soils' microbiomes. The EN soil communities are represented by a wide variety of bacteria and fungi. The presented soil parameters can be used as a reference in assessing the increasing impact of fires on pine forests and podzols.

**Keywords:** boreal forest; podzol; PAHs; pyrogenic history; microbial biomass; prokaryotic; fungal communities

## 1. Introduction

The albic podzols are among the most recognizable soils in the world. The most common vegetation growing on podzols is pine (*Pinus sylvestris* L.) forests. Podzols occupy about 7% of the territory of the Russian Federation. Podzols are often considered as azonal soil types, mainly confined to the boreal and tundra zones. On the territory of Russia, podzols are widespread both in the European North and in the Asian part. Significant areas of podzols are represented in boreal forest regions. In the Komi Republic, they are distributed on 10% of the territory (41,590 km$^2$) and in the Krasnoyarsk region, on 3.7% (86,568 km$^2$) [1]. Podzols are characterized by a well-defined morphological structure of



the profile. The most general structure of the profile is as follows: O-E-Bs-BC. Podzols are characterized by low biological productivity and low natural fertility. They are rarely used in agriculture due to their low natural fertility. In the boreal zone, the territories occupied by podzols are mainly of forestry importance. One of the most common factors affecting these ecosystems is fire [2]. Fires in boreal forests, particularly in pine forests, are a natural historical factor in their development [3–7]. Among the pine forests, lichen and dwarf types, growing on sandy and sandy loam soils, are most vulnerable in terms of natural fire hazards [8,9]. The sandy texture of the soil-forming rocks contributes to good drainage, with the soils and litter drying out, and provides the highest fire hazard [10]. Recent forest fires are an important factor in the existence of both pine forests and albic podzols [11,12]. Pyrogenic signs at the morhpones and buried horizons of podzols can last up to 5000 years [13]. Podzols under pine forests are the most pyrogenic soils on the planet and their morphological and physicochemical properties can be determined by the age of fires.

A number of recent studies revealed that fire can affect the soil organic matter (SOM). Part of pyrogenically changed carbon compounds are stable enough [14,15] and play an important role in the global carbon cycle [16–18]. Santin and Doerr [14] showed that soils accumulate significant pools of pyrogenic carbon (PyC). Changes in soil organic matter during fires are often diagnosed by the content of compounds of an aromatic nature, in particular benzene polycarboxylic acids (BPCAs) or polycyclic aromatic hydrocarbons (PAHs) [19–24]. According to Rey-Salgueiro et al. [25], the greatest PAH amount is formed at temperatures of 200–400 °C, which are more typical for surface fires in the taiga zone. PAHs are normally formed during biomass burning [26]. PAHs are known to have toxic, cancerogenic, and mutagenic properties [27].

Despite the wide distribution of podzols, there is no comparison of the properties of soils in different geographic regions. The main purpose of this study was to conduct a comprehensive assessment of the soils of post-pyrogenic old-age pine forests of the European North and Central Siberia, taking into account their pyrogenic history. This work is based on the identification of soil features, organic matter, and microbiological properties associated with the influence of an old pyrogenic impact. We hypothesized that (1) the basic chemical properties of the soils of the studied regions are close to each other, (2) soils with a similar pyrogenic history of forests may show differences in the intensity of fires and accumulation of PAHs, and (3) the microbiological activity in soils after fire decreases and does not reach the optimal values for a long time. The identification of local geographic features and the assessment of dominant properties are essential to understanding the development and evolution of these soils. The objectives of this study were to compare the fire history of pine forests in EN and CS; (ii) characterize and compare the morphological, chemical, and microbiological properties of podzols; and (iii) identify the features of the microbiological characteristics of soils associated with the influence of the last fire.

## 2. Materials and Methods

### 2.1. Area Description and Soil Sampling

This paper presents data on six soil pits formed under lichen pine forests (Supplementary Figure S1). Three soils (109 EN, 113 EN, 131 EN) formed on the territory of the Komi Republic and the European North (EN) (the territory of the Pechora-Ilych nature reserve 61°44′−61°59′ N, 57°06′−58°08′ E), and three soils (45 CS, 79 CS, 121 CS) on the territory of Central Siberia (CS) and the Krasnoyarsk region (the vicinity of the ZOTTO International Observatory 60°44′−60°48′ N, 88°48′−88°26′ E) were studied (Figure 1). The number before the abbreviation means how many years have passed since the last fire. Samples were taken from each genetic horizon of the studied soils. Soils were classified according to IUSS [28]. We added «pyr» to the horizon name when inclusions associated with fires were diagnosed (mostly coal and soot residues) during the field description [29–32].

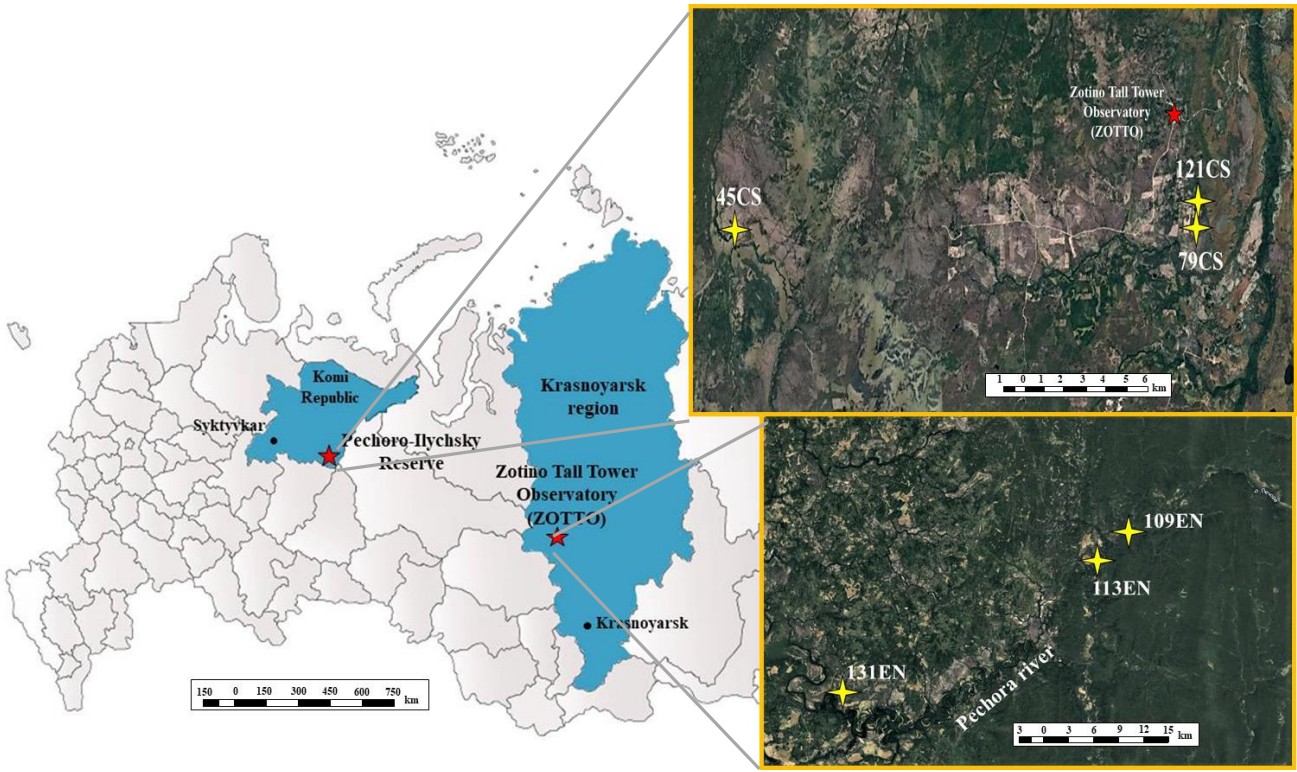

**Figure 1.** Location of study sites (the scheme is based on https://glovis.usgs.gov/, accessed on 25 May 2022).

### 2.2. Geobotanical and Dendrochronological Studies

Reveles (vegetation descriptions) were made were made at 400 m$^2$ sites using the standard geobotanical methods [33]. For the tree layer, we described its composition. Undergrowth, herb-dwarf shrub, and moss-lichen layers were characterized by the relative abundance of species and the total projective cover (TPC) of each layer. Latin names were given according to www.worldfloraonline.org/, accessed on 25 May 2022.

Tree-ring samples for the estimation of the fire scars were obtained according to Madany et al. [34]. About 50–100 cores and 3–7 (cuts) cross-sections were taken from live trees on each plot. The preparation of wood samples (cores and cross-sections) for the dating of fires was carried out according to Fritts [35] and Grissino-Mayer [36]. The tree ring width was measured with an accuracy of 0.01 mm on a LINTAB semiautomatic device under a binocular microscope with a 40× magnification. Using TSAPW in the software environment with cross-dating (Cross-Dating), the calendar year of each ring and fire scar was determined [37]. For dating accuracy, the data collected were checked using cross-correlation analysis using the COFECHA software [38,39].

### 2.3. General Soil Analysis

For chemical analysis, mixed samples (from five points) within an area of approximately 2 m$^2$ were collected. Samples were air-dried, and living visible roots and all particles with a diameter >2 mm were removed by dry-sieving as preparation for the analysis. Organic samples were ground with a sample grinder WCG75 (Pro Prep, Torrington, CT, USA).

Chemical analyses of the soils were performed in the Ecoanalit laboratory and Soil science department of the Institute of Biology, Federal Research Center, Komi Research Center of the Ural Branch of the Russian Academy of Sciences (IB FRC Komi SC UrB RAS). The pH values were determined on an HI2002-02 edge series pH meter with a Hanna digital pH electrode (Hanna Instruments, Nusfalau, Romania) at a soil:water ratio of 1 (m):2.5 (v) for mineral horizons and 1 (m):25 (v) for organic and pyrogenic horizons, respectively.

Exchangeable cations ($Ca^{2+}$, $Mg^{2+}$, $K^+$, $Na^+$) were extracted 1M CH3COONH$_4$ on a mechanical extractor (Sampletek, Mavco Industries Inc., Lawrenceburg, TN, USA) according to Van Reeuwijk [40]. The cation concentrations were detected by inductively coupled plasma atomic emission spectroscopy (ICP Spectro Ciros CCD, Spectro Al, Kleve, Germany). The cation exchange capacity (CEC) was determined by saturation with $Na^+$ ions according to [40]. Base saturation (BS) was defined as the ratio of the sum of exchangeable cations ($\sum$) to CEC multiplied by 100%. The texture of soils was determined according to Van Reeuwijk [40]. The classification of the soil texture was determined using the Ferre triangle. The specific surface area (SSA) of the samples were determined on a surface area analyzer (Sorbtometr-M, Katakon, Novosibirsk, Russia) with $N_2$ as the adsorbate gas at the Faculty of Soil Science, Lomonosov Moscow State University (Moscow, Russia). This method was described in detail earlier [14].

### 2.4. Soil Organic Matter, Water-Soluble Organic Carbon and Nitrogen, and PAH Contents

$C_{tot.}$ and $N_{tot.}$ were determined by dry combustion on an EA-1100 analyzer (Carlo Erba, Milano, Italy). Water-soluble organic carbon (WSOC) and nitrogen (WSON) were extracted with deionized water (ELGA Lab Water, England) at room temperature at a ratio of 1:50 (soil:water) for mineral horizons and 1:100 for organic horizons in BIOFIL test tubes. WSOC and WSON were assessed using the TOC-VCPN analyzer (Shimadzu, Kyoto, Japan) with the TNM-1 module. The method was described in detail earlier [41].

The extraction of PAHs from the soils was carried out on an Accelerated Solvent Extractor 350 (Thermo Scientific™, Waltham, MA, USA) in the Chromatography Collective Use Center of the IB FRC Komi SC UrB RAS. The US EPA method 8310 (1986) and certified national standard method of quantitative chemical analysis (PND F 16.1:2.2:2.3:3.62-09, 2009) were selected for PAH measurement. The content of PAHs in the concentrates was determined by reversed-phase high-performance liquid chromatography in gradient mode with spectrofluorimetric detection. The following PAHs were assessed: naphthalene (NP), fluorene (FL), phenanthrene (PHE), anthracene (ANT), fluoranthene (FLA), pyrene (PYR), benzo[a]anthracene (BaA), chrysene (CHR), benzo[b]fluoranthene (BbF), benzo[k]fluoranthene (BkF), benzo[a]pyrene (BaP), dibenzo[a,h]anthracene (DahA), benzo[g,h,i]perylene (BghiP), and indeno [1,2,3-c,d]pyrene (IcdP). The analysis technique was described in detail earlier [42].

### 2.5. Microbiological Parameters

We estimated same soil-microbiological indicators of the respiratory activity of soils: basal respiration (BR), microbial biomass (MB (Cmic)), and the microbial metabolic coefficient (QR) [43,44]. The samples were placed in a freezer at a temperature of minus 20 °C until the start of the analysis. The respiratory activity of the peat soil microbial community was expressed by BR and substrate-induced respiration (SIR) using an Agilent Technologies 6890 N Network GC gas chromatograph (USA). The SIR values were used to calculate the carbon of the microbial biomass of the microorganisms (Cmic = MB). QR was calculated as the ratio of BR to SIR, and the contribution of microbial carbon ($C_{mic}$, %) to the total organic C as the ratio $C_{mic}/C_{tot.}$ [43–45]. Detailed methods were given in the articles by Grodnitskaya et al. [46,47].

Molecular genetic analysis of the soil samples was performed at the Research Institute of Agricultural Microbiology (Pushkin, St. Petersburg, Russia). A set of reagents (NucleoSpin Soil) from MACHEREY-NAGEL (Düren, Germany) was used to isolate DNA from the soil samples according to the manufacturer's instructions. The taxonomic composition of the fungal, bacterial, and archaeal communities was determined based on analysis of the amplicon libraries of the ribosomal operon fragments in each sample.

Taxonomic analysis of the fungal community was determined based on analysis of the amplicon libraries of the fungal ribosomal operon fragments (ITS2) obtained by PCR using ITS1/ITS2 primers (GCATCGATGAAGAACGCAGC/TCCTCCGCTTATTGATATGC). Taxonomic analysis of the bacterial community was carried out with universal primers

F515/R806 for the variable region of the 16SrRNA v4 gene (GTGCCAGCMGCCGCG-GTAA/GGACTACVSGGGTATCTAAT) specific for a wide range of microorganisms, including bacteria [48]. Taxonomic analysis of archaea was performed with the primers A956F/A1401R (TYAATYGGANTCAACRCC/CRGTGWGTRCAAGGRGCA) [49]. All primers had service sequences containing linkers and index sequences (required for Illumina sequencing).

For different sets of primers, the same PCR program was chosen: the reaction mixture (15 μL) contained 0.5 units of activity of Q5® High-Fidelity DNA Polymerase (NEB, Ipswich, MA, USA), 5 pM each of forward and reverse primers, 10 ng of DNA template, and 2nM of each dNTP (Life Technologies, Carlsbad, CA, USA). The mixture was denatured at 94 °C for 1 min, followed by 25 cycles: 94 °C for 30 s, 50 °C for 30 s, and 72 °C for 30 s. The final elongation was carried out at 72 °C for 3 min. PCR products were purified according to the Illumina-recommended method using AMPureXP (BeckmanCoulter, Brea, CA, USA). Further preparation of the libraries was carried out in accordance with the manufacturer's MiSeq Reagent Kit Preparation Guide (Illumina, San Diego, CA, USA) (https://support.illumina.com/documents/documentation/chemistry_documentation/16s/16s-metagenomic-library-prep-guide-15044223-b.pdf, accessed on 25 May 2022). The libraries were sequenced according to the manufacturer's instructions on an Illumina MiSeq instrument (Illumina, San Diego, CA, USA) using a MiSeq® ReagentKit v3 (600 cycle) with double-sided reading (2 × 300 *n*). Initial data processing, namely, sample demultiplexing and adapter removal, was carried out by Illumina software (Illumina, USA). For subsequent denoising, sequence merging, restoration of original phylotypes (ASV, Amplicon sequence variant), removal of chimeric reads, and further taxonomic classification of the resulting ASVs (using the SILVA database for prokaryotes and Unite for fungi (release 132 [50])), the software packages dada2 [51], phyloseq [52], and DECIPHER [53] were used, which were run in the R software environment. The QIIME software package was used [54].

### 2.6. Statistics

Correlation analysis was used to determine the relationship between the obtained data. Correlation coefficients (r-Pearson) were calculated using STATISTICA 10.0 (Stat. Soft Inc, Tusla, OK, USA); differences were considered significant at the significance level $p < 0.05$. For PCA, the Past 3.0 program was used.

## 3. Results

### 3.1. Vegetation at the Study Sites and Pyrogenic History in the Study Areas

At all study sites, vegetation cover is presented as pine forests (*Pinus sylvestris* L.) with the lichen type, with a predominance of dwarf shrubs (*Vaccinium vitis-idaea*, *Ledum palustre*) and lichens (*Cladonia arbuscular*, *Cladonia rangiferina*, *Cladonia crispata*) in the lower vegetation layers (Table 1). The plant communities have similar qualitative features (floristic composition and structure) and were classified as *Pinetum vaccinioso-cladinosum* plant association in both the Komi and Krasnoyarsk regions.

The analysis of fire activity in the studied areas (Table 2) showed that at the time of their development, the forests were exposed to fires four to five times. The fire return interval (FRI) was 5 to 103 years, with an average burning period of 26 to 59 years. In the territory of the Komi Republic, the surveyed areas are characterized by an inter-fire period of 5 to 82 years, with an average fire frequency of 27 to 45 years. Forests in the Zotino area are characterized by an inter-fire interval of 14 to 103 years, with an average fire frequency of 26 to 59 years. The highest fire activity in all areas manifested in the XVII, XIX, and early XX centuries.

**Table 1.** Total projective cover (TCP) and dominant species of the lower layers of plant communities of *Pinetum vaccinioso-cladinosum* formed on albic podzols.

| Site | Total Projective Cover (TCP) and Dominant Species of Lower Layers of Plant Communities |
|---|---|
| | Zotino (Krasnoyarsk region), Central Siberia (CS) |
| 45 CS | Dwarf-shrub herb layer (TPC 30%), <br> *Vaccinium vitis-idaea, Ledum palustre, Vaccinium uliginosum* <br> Moss-lichen layer (TPC 90%), <br> *Cladonia arbuscula, Cladonia rangiferina, Cladonia crispata, Pleurozium schreberi* |
| 79 CS | Dwarf-shrub herb layer (TPC 25%), <br> *Vaccinium vitis-idaea. Calamagrostis obtusata, Diphasium complanatum, Vaccinium myrtillus* <br> Moss-lichen layer (TPC 80%), <br> *Cladonia arbuscula, Cladonia rangiferina, Cladonia crispata, Pleurozium schreberi* |
| 121 CS | Dwarf-shrub herb layer (TPC 5%), <br> *Vaccinium myrtillus, Vaccinium vitis-idaea, Ledum palustre, Diphasium complanatum, Empetrum hermaphroditum* <br> Moss-lichen layer (TPC 90%), <br> *Cladonia arbuscula, Cladonia rangiferina* |
| | Pechora-Ilychsky nature reserve (Komi Republic), European North (EN) |
| 109 EN | Dwarf-shrub herb layer (TPC 30%–40%), <br> *Vaccinium vitis-idaea, Vaccinium myrtillus, Calamagrostis purpurea, Avenella flexuosa, Diphasiastrum complanatum* <br> Moss-lichen layer (TPC 80%–90%), <br> *Cladonia stellaris, Cladonia rangiferina, Cladonia arbuscula, Pleurozium schreberi, Ptilium crista-castrensis* |
| 113 EN | Dwarf-shrub herb layer (TPC 40%–50%), <br> *Vaccinium vitis-idaea, Vaccinium myrtillus, Diphasiastrum complanatum* <br> Moss-lichen layer (TPC 80%–90%), <br> *Cladonia stellaris, Cladonia rangiferina, Cladonia arbuscula, Pleurozium schreberi, Ptilium crista-castrensis* |
| 131 EN | Dwarf-shrub herb layer (TPC 50%–60%), <br> *Vaccinium vitis-idaea* <br> Moss-lichen layer (TPC 80%), <br> *Cladonia stellaris, Cladonia rangiferina, Cladonia arbuscula, Pleurozium schreberi, Polytrichum commune* |

Abbreviation: CS—Central Siberia, EN—European North. The number before the abbreviation means how many years have passed since the last fire.

**Table 2.** Dates of fires with a fire period for the objects under study.

| Site | Dates of Fires | | | | | Accounting Year | Fire Interval, Years | | | | Average Frequency of Fires, Years | The Last Fire, Years |
|---|---|---|---|---|---|---|---|---|---|---|---|---|
| | 1 | 2 | 3 | 4 | 5 | | 1–2 | 2–3 | 3–4 | 4–5 | | |
| Central Siberia (CS) | | | | | | | | | | | | |
| 45 CS | 1974 | 1953 | 1939 | 1911 | 1869 | 2019 | 21 | 14 | 28 | 42 | 26 | 45 |
| 79 CS | 1940 | 1898 | 1873 | – | – | 2019 | 42 | 25 | – | – | 34 | 79 |
| 121 CS | 1898 | 1825 | 1792 | 1766 | 1663 | 2019 | 73 | 33 | 26 | 103 | 59 | 121 |
| European North (EN) | | | | | | | | | | | | |
| 109 EN | 1910 | 1904 | 1880 | 1830 | – | 2019 | 6 | 24 | 50 | – | 27 | 109 |
| 113 EN | 1911 | 1906 | 1869 | 1841 | 1772 | 2019 | 5 | 37 | 28 | 69 | 45 | 113 |
| 131 EN | 1887 | 1805 | 1785 | 1744 | 1727 | 2018 | 82 | 20 | 41 | 17 | 40 | 131 |

Site designations are presented in Table 1.

### 3.2. Morphological Properties of Soils

The studied podzols are characterized by a typical morphology. The structure of the soil profile of the soils is: $Oi/Q_{pyr}$—$O_{e,pyr}$—$E_{pyr}$—E—Bs—BC. The organic horizon consists of several subhorizons with different decomposition of plant organic material. The upper subhorizon Oi consists of slightly decomposed lichens, moss remnants, branches, bark, and pine needles. The $O_{e,pyr}$ subhorizon is a medium-decomposed organic material with

an abundance of charcoals of various sizes. Below, a light gray (10 YR5/2) sandy horizon $E_{pyr}$ is formed. At the top, $E_{pyr}$ is impregnated with organic matter. Horizon E is whitish (7.5 YR 8/1), without structures, sometimes with charcoal. Horizon Bs formed below. Horizon Bs is light brown (7.5 YR6/8), sand, without a structure, and is characterized by the accumulation of iron compounds. The lowest horizon BC has a non-uniform color (2.5.6/6–2.5 Y 7/4–10 YR 4/6). This horizon also has no structures.

The thickness and moisture content of the organic horizons are important characteristics of soils (Figure 2). The highest values of the thickness of the organic horizon were found for the soil of the site 113 years (113 EN) after the fire (10.7 ± 1.5 cm). The minimum moisture content of the organic horizons was found in the area of 45 CS. At the site 79 CS, an abnormally low moisture content level (22%) was detected. The moisture content of the litter in the pyrogenic areas of a later age increased with the restoration of the ground cover and varied from 98 to 152%.

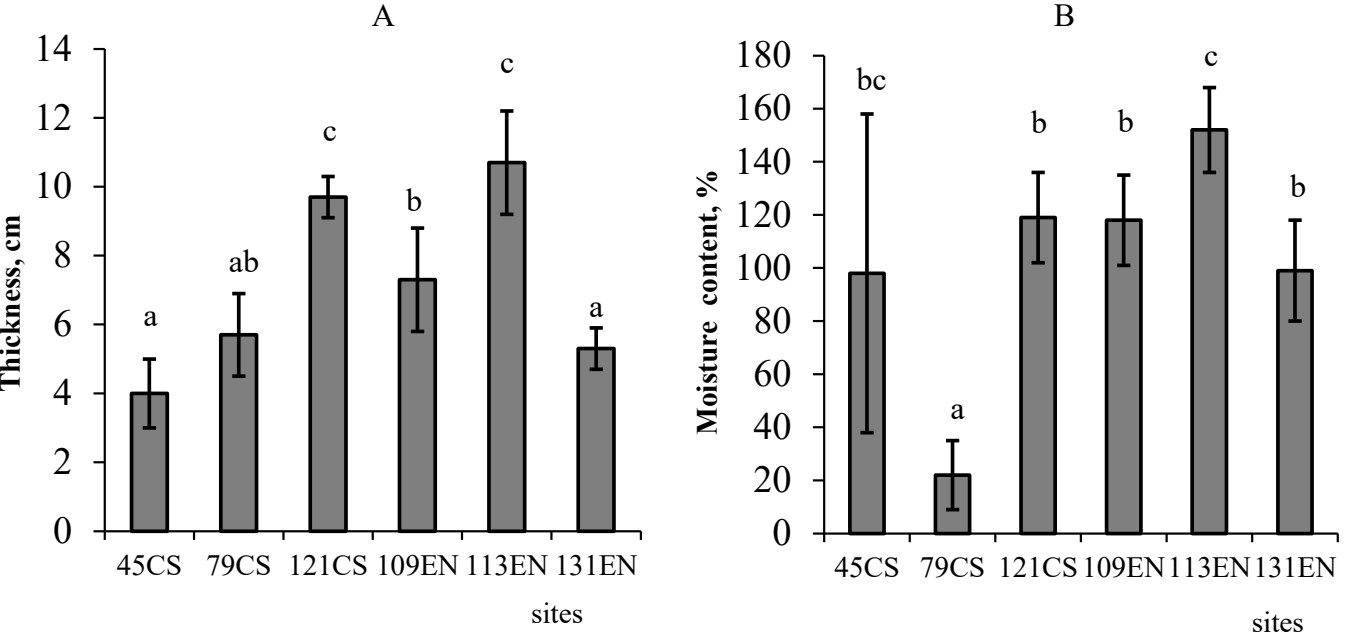

**Figure 2.** The thickness of the organic horizon (**A**) and moisture content (**B**) in the study soils ($n = 3$, the error bars are the standard errors). Abbreviation: CS—Central Siberia, EN—European North. The number before the abbreviation means how many years have passed since the last fire. Different lowercase letters indicate significant differences at $p < 0.05$.

### 3.3. Physicochemical Properties of Soils

The soils are strongly acidic. The $pH_{H2O}$ value varied from 4.1 to 4.8 in the organic horizons (Figure 3A). In the mineral horizons, the $pH_{H2O}$ value ranged from 4.7 to 6.1. The most common trend of the profile changes in acidity is an increase in the pH values from the upper mineral horizons to the lower ones.

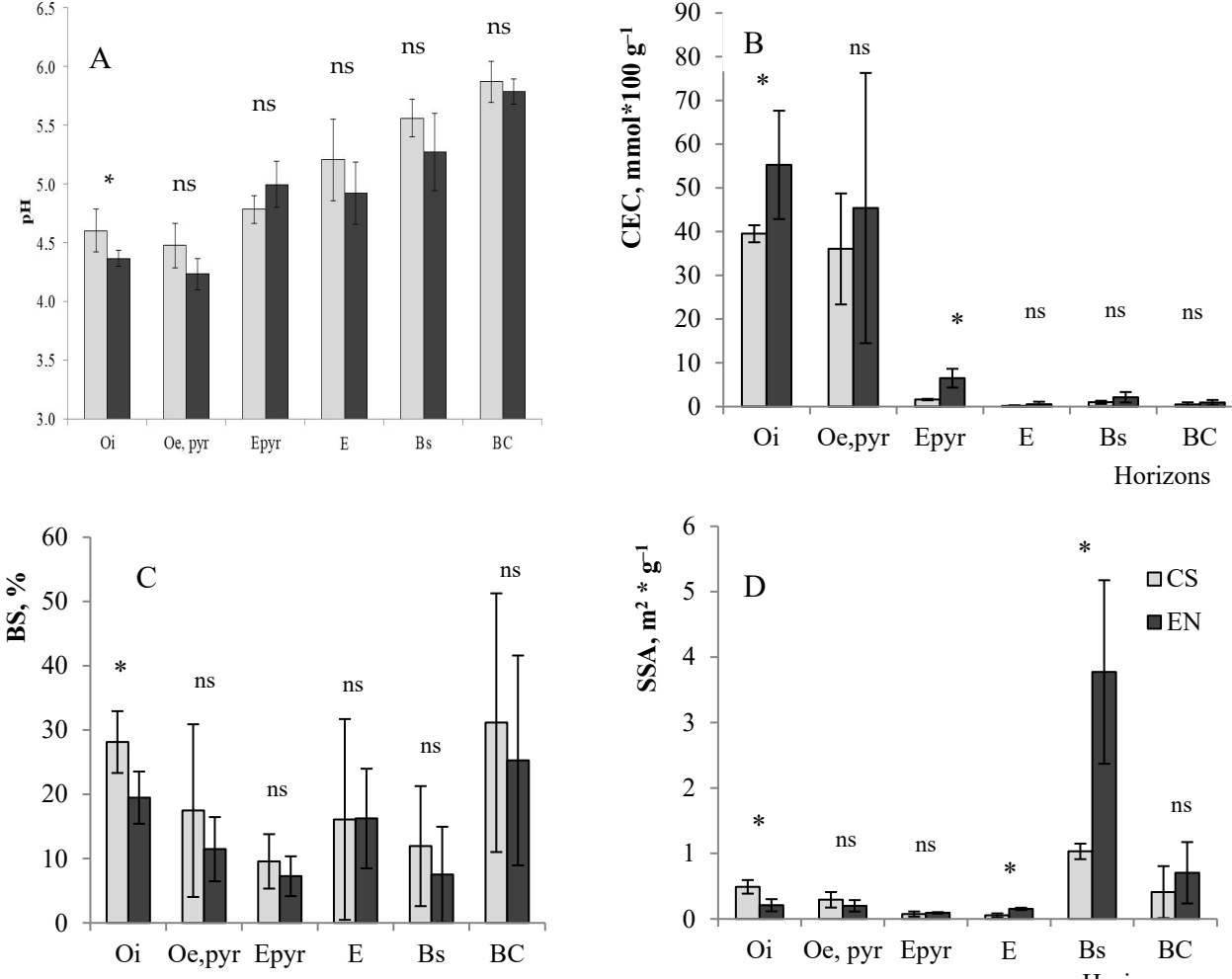

**Figure 3.** pH (water) values (**A**), cation exchange capacity (**B**), and base saturation (**C**) and specific surface area (SSA) values for albic podzols (**D**) (*n* = 3, the error bars are the standard errors). Abbreviation: CS—Central Siberia, EN—European North. *—indicate significant differences between EN and CS, ns—differences between EN and CS are not significant at *p* < 0.05.

The analysis of the texture composition (Supplementary Table S1) showed that the studied soils are characterized by a weak profile differentiation by granulometric fractions. It was revealed that these soils are characterized by a low content of silty fraction (1%–42%). The content of the sand fraction prevails and varies from 14 to 98% while the content of the clay fraction is 0%–44%. The mineral horizons of the studied soils belong to different types of sands. There was no direct pyrogenic effect on the content of individual granulometric fractions. The differences in the compared sites are probably determined by the initial lithological heterogeneity.

The organic horizons are characterized by the maximum content of exchange cations (Figure 3B,C). A significant decrease in the content of exchange cations was found in the mineral horizons. Among the exchange bases, calcium cations predominate. The content of $Ca^{2+}$ varies from 0 to 10.3 mmol 100 $g^{-1}$. The content of exchangeable $Mg^{2+}$ is 0–3.1 mmol 100 $g^{-1}$ of the soil. The content of exchangeable $K^+$ cations in soils varies from 0.001 to 2.6 mmol 100 $g^{-1}$. The minimum content was found for $Na^+$ cations (from 0 to 0.44 mmol 100 $g^{-1}$). In addition, an increase in the exchange cations in the lower mineral horizons was revealed. The sum of the exchange cations in the lower mineral horizons ranges from 0.09 to 6.48 mmol 100 $g^{-1}$.

SSA of sandy soils is characterized by very low values (Figure 3D). The minimum values were found for the E and $E_{pyr}$ horizons. The SSA values of the Bs horizons are

significantly higher. SSA of the illuvial horizons of the EN podzols is significantly higher than that of the illuvial horizons of soils in Central Siberia.

### 3.4. Carbon and Nitrogen Contents

The highest concentrations of carbon and nitrogen were found in the organic soil horizons (Figure 4A,B). The $C_{tot.}$ and $N_{tot.}$ contents were decreased in the mineral horizons. The $C_{tot.}$ concentrations in the studied soils vary from 1.0 to 497 g kg$^{-1}$ and the nitrogen concentrations from 0.1 to 9.4 g kg$^{-1}$. The key characteristic of the postpyrogenic soils is the carbon content in the upper mineral horizon. We found an increase in the content of carbon (11.8–80 g kg$^{-1}$) and nitrogen (0.6–2.1 g kg$^{-1}$) in the pyrogenic horizons ($E_{pyr}$). The maximum content was found at $E_{pyr}$ of the soil of the area affected by fire 109 years ago (80 g kg$^{-1}$). The same area is characterized by the highest frequency of fires.

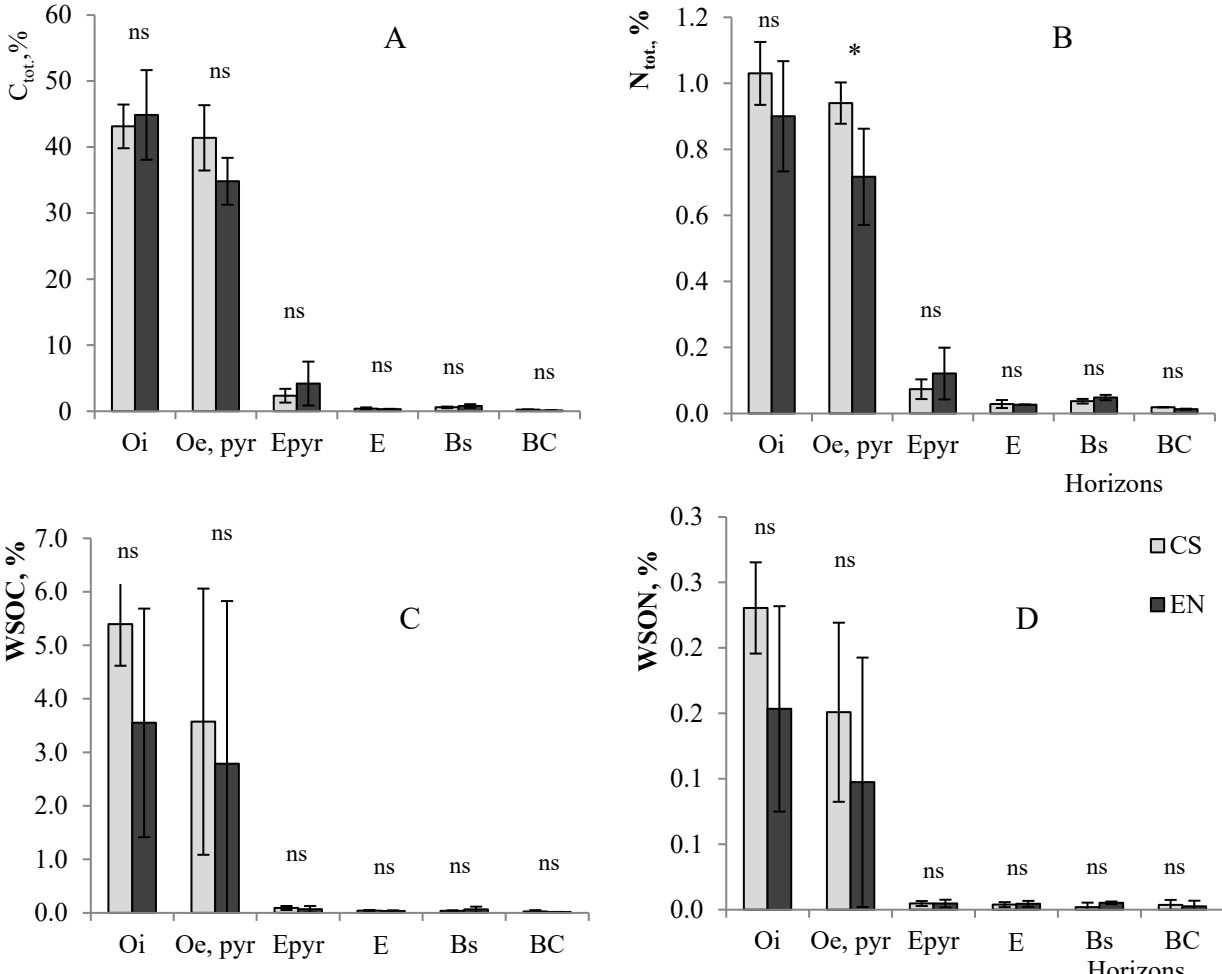

**Figure 4.** Carbon (**A**) and nitrogen (**B**), water-soluble carbon compound, (**C**) and water-soluble nitrogen compound (**D**) contents at horizons of studied soils ($n = 3$, the error bars are the standard errors). Abbreviation: CS—Central Siberia, EN—European North. *—indicate significant differences between EN and CS, ns—differences between EN and CS are not significant at $p < 0.05$.

The organic horizons were characterized by the maximum concentrations of WSOC and WSON of the studied soils. The mineral horizons contained lower WSOC and WSON (Figure 4C,D). Therefore, we only considered the organic and upper mineral horizons. The WSOC content of the 131 year site in the organic horizon is 3.2–4.6 mg g$^{-1}$ and the WSON content is 0.17–0.19 mg g$^{-1}$. In the upper mineral horizon, the WSOC content is 0.082 mg g$^{-1}$.

At sites 45 years after the fire (45CS), high values of WSOC and WSON were observed in the organic horizons. Similarly, there was an increase in water-soluble organic compounds in the $E_{pyr}$ horizon in the range from 0.08 to 0.10 mg g$^{-1}$. In the areas affected by fires 79 (79 CS), 109 (109 EN), and 131 (131 EN) years ago, smaller and similar values of the carbon content WSOC and nitrogen WSON were found. Carbon values ranged from 1.1 to 2.6 mg g$^{-1}$. The nitrogen content was 0.04–0.14 mg g$^{-1}$. Similar values were also found for the upper mineral horizons: 0.03–0.12 mg g$^{-1}$. The maximum content of water-soluble organic compounds was found for the site that was exposed to fire 131 years ago (131 EN): 6.2–6.5 (WSOC) and 0.20–0.24 mg g$^{-1}$ (WSON). WSOC of 0.16 mg g$^{-1}$ is also maximum in the $E_{pyr}$ horizon. The share of water-soluble carbon from the total carbon content in the soil profile is uneven. In the lower mineral horizons, it was found to increase up to 2.5%.

### 3.5. PAH Contents

The PAH contents was determined in the organic and upper mineral ($E_{pyr}$) horizon (Table 3). Only the upper horizons were studied because the highest concentrations of the compounds under consideration are observed here [11]. All horizons are characterized by a high content of PAHs. Polyarenes in organic soil horizons are mainly represented by light structures. The contribution of 5–6-nuclear structures to the total mass fraction of PAHs in organic horizons ranged from 5%–6% in areas recently burned to 20% in older burning sites and naturally increases with age. Correlation analysis of the obtained data of the $E_{pyr}$ horizon did not reveal significant relationships between the number of fires and the frequency of fires with chemical indicators. However, there is a certain tendency for some PAHs to depend on the age after a fire: naphthalene (r = 0,89, $p < 0.05$), phenanthrene (r = 0.58, $p < 0.05$), anthracene (r = 0.63, $p < 0.05$) and benzo[k]fluoranthene (r = 0.64, $p < 0.05$). The content of total C and N is significantly correlated with 4-, 5-, and 6-nuclear PAHs. The correlation coefficients are r = 0.85–0.95 for carbon and r = 0.85–0.94 for nitrogen. Dibenzo[a,h]anthracene and benzo[g,h,i]perylene are significantly correlated with the carbon stocks in the pyrogenic horizon $E_{pyr}$ (r = 0.85, $p < 0.05$). In addition, PAHs have high values for the correlation coefficient with CEC (r = 0.85–0.99, $p < 0.05$). The granulometric fractions of fine/very fine sand and coarse silt of the upper mineral horizon have high values of the correlation coefficient with the content of individual PAHs (r = 0.82–098, $p < 0.05$) and their sum ($\sum$PAHs) (r = 0.81–0.87, $p < 0.05$). Nevertheless, significant negative values of the correlation coefficient of the PAH content and the medium sand fraction were revealed (r = $-0.86 \ldots -0.98$, $p < 0.05$).

In general, the patterns of formation of the PAH composition of organic horizons of soils of the Komi Republic and the Krasnoyarsk region are similar. The content of PAHs in the organic horizon of soils can be arranged in a row: 113, 121, and 133 years after the fire.

The mass fraction of both light and heavy structures decreased with an increase in the time elapsed since the last fire. At the same time, there were differences in the composition of PAHs. Significant amounts of indenopyrene were found in the organic soil horizons of EN, which were not detected in other sites (Figure 5). Heavy PAHs in insignificant amounts were only present in the mineral horizons of the soils of EN.

**Table 3.** PAH content in the upper soil horizons (x ± δ), ng g$^{-1}$.

| Site | Horizon | Depth, cm | 2-Ring | | 3-Ring | | 4-Ring | | | | 5-Ring | | | | 6-Ring | | ΣPAHs |
|---|---|---|---|---|---|---|---|---|---|---|---|---|---|---|---|---|---|
| | | | NP | FL | PHE | ANT | FLA | PYR | BaA | CHR | BbF | BkF | BaP | DahA | BghiP | IcdP | |
| | | | | | | | | | Zotino (Krasnoyarsk region), Central Siberia (CS) | | | | | | | | |
| 45 CS | Oe,pyr | 3–5 | 127 ± 64 | 8 ± 3 | 76 ± 38 | 4 ± 2 | 17 ± 8 | 5 ± 2.3 | 2 ± 0.8 | 9.2 ± 5 | 7.3 ± 3 | 2 ± 1 | 5.3 ± 3 | 3.4 ± 1.6 | – | – | 266.1 |
| | Epyr | 5–11 | 6.5 ± 3.2 | 1.3 ± 0.5 | 6.7 ± 3.3 | 0.4 ± 0.2 | 2.6 ± 1.2 | 0.8 ± 0.4 | 0.3 ± 0.1 | 1 ± 0.5 | – | – | – | – | – | – | 19.6 |
| 79 CS | Oe,pyr | 1–2 | 75 ± 37 | 7 ± 3 | 121 ± 27 | 5 ± 3 | 9.5 ± 4.4 | 5.4 ± 2.5 | 1.5 ± 0.6 | 2.5 ± 1.3 | 6.1 ± 2.6 | 3.2 ± 1.5 | 3.6 ± 1.8 | 15 ± 7 | 1.8 ± 0.8 | – | 256.6 |
| | Epyr | 2–6 | 8.9 ± 4.4 | 1.2 ± 0.5 | 6.8 ± 3.4 | 0.3 ± 0.1 | 3.5 ± 1.6 | 2.1 ± 0.9 | – | 1.1 ± 0.5 | – | – | – | – | – | – | 23.8 |
| 121 CS | Oe,pyr | 2–4 | 191 ± 96 | 9 ± 4 | 95 ± 21 | 3.8 ± 2 | 9 ± 4 | 4 ± 1.8 | 0.8 ± 0.3 | 23 ± 12 | 62 ± 26 | 0.9 ± 0.4 | 5.8 ± 2.9 | 29 ± 14 | – | – | 432.7 |
| | Epyr | 4–10 | 13 ± 7 | 0.7 ± 0.3 | 7.5 ± 3.7 | 1 ± 0.5 | 1.7 ± 0.8 | 0.7 ± 0.3 | 0.4 ± 0.2 | 2.2 ± 1.1 | – | – | – | – | – | – | 27.5 |
| | | | | | | | | | Pechora-Ilychsky nature reserve (Komi Republic). European North (EN) | | | | | | | | |
| 109 EN | Oe,pyr | 2–4 | 184 ± 92 | 13 ± 5 | 161 ± 35 | 4.2 ± 2 | 22 ± 10 | 5.8 ± 2.7 | 2.6 ± 1.1 | 31 ± 16 | 14 ± 6 | 3.2 ± 1.5 | 8.7 ± 4.3 | 14 ± 6 | 3.9 ± 1.7 | 18 ± 10 | 483.0 |
| | Epyr | 4–7 | 16 ± 8 | 3.6 ± 1.4 | 40 ± 20 | 3.1 ± 1.6 | 5.8 ± 2.7 | 7.2 ± 3.3 | 1.3 ± 0.5 | 7.4 ± 3.8 | 5.2 ± 2.1 | 0.3 ± 0.1 | 2.7 ± 1.3 | 3.8 ± 1.8 | 0.7 ± 0.3 | – | 97.0 |
| 113 EN | Oe,pyr | 2–4 | 130 ± 65 | 16 ± 6 | 272 ± 60 | 7.2 ± 3.6 | 31 ± 14 | 18.5 ± 8.5 | 2.3 ± 1.0 | 27 ± 14 | 47 ± 20 | 4.9 ± 2.4 | 7.2 ± 3.6 | 14 ± 7 | 4.6 ± 2.0 | 41 ± 22 | 622.9 |
| | Epyr | 4–7 | 13 ± 7 | 2.9 ± 1.2 | 25 ± 12 | 1.5 ± 0.7 | 4.9 ± 2.3 | 4 ± 2 | 0.4 ± 0.2 | 0.9 ± 0.5 | – | 0.2 ± 0.1 | 0.4 ± 0.2 | – | – | – | 53.4 |
| 131 EN | Oe,pyr | 1–3 | 61 ± 31 | 3.7 ± 1.5 | 159 ± 35 | 5 ± 3 | 18 ± 8 | 2.6 ± 1.2 | 1.3 ± 0.5 | 7 ± 4 | 49 ± 20 | 2.6 ± 1.1 | 5.2 ± 2.6 | – | 1.6 ± 0.7 | – | 315.8 |
| | Epyr | 3–7 | 21 ± 10 | 3.4 ± 1.4 | 32 ± 16 | 1.7 ± 1 | 4.7 ± 2.2 | 4.7 ± 2.2 | 0.7 ± 0.3 | 3.7 ± 1.9 | 3 ± 1.3 | 0.5 ± 0.2 | 1.3 ± 1 | – | – | – | 76.6 |

Note: NP naphthalene, FL fluorene, PHE phenanthrene, ANT anthracene, FLA fluoranthene, PYR pyrene, BaA benzo[a]anthracene, CHR chrysene, BbF benzo[b]fluoranthene, BkF benzo[k]fluoranthene, BaP benzo[a]pyrene, DahA dibenzo[a,h]anthracene, BghiP benzo[g,h,i]perylene, IcdP indeno[1,2,3-c,d]pyrene. Dash—below the limit of determination. Designations of study sites are the same as in Table 1.

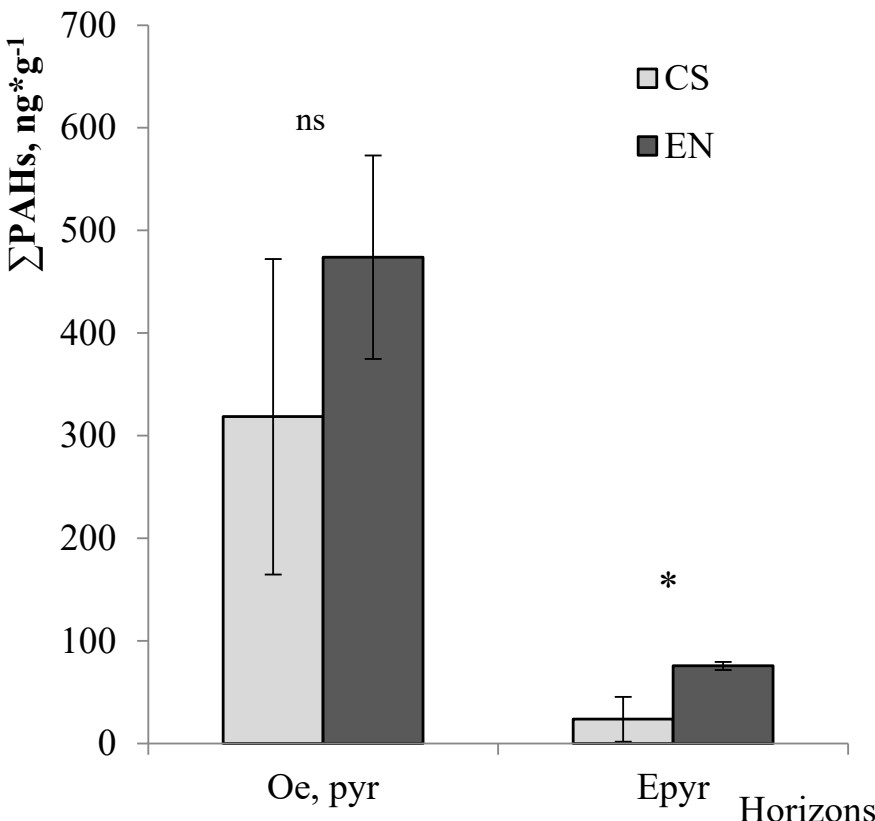

**Figure 5.** The total content of PAHs in the upper horizons of the studied soils (*n* = 3, the error bars are the standard errors). Abbreviation: CS—Central Siberia, EN—European North. *—indicate significant differences between EN and CS, ns—differences between EN and CS are not significant at $p < 0.05$.

### 3.6. Microbiological Properties

The podzols of CS and EN showed a generally similar trend in the distribution of MB, BR, and QR along the horizons, with a difference in their quantitative content. The MB and BR values decreased down the profile and varied depending on the organic matter content.

Podzols are characterized by a low MB content in the mineral soil horizons and high values of MB and BR in the organic horizons (Figure 6). The microbiological parameters (MB, BR, and QR) in the $E_{pyr}$ horizon are the most variable. The upper litter subhorizons (Oi) are characterized by the highest biomass. The MB and BR values vary from 978 to 1440 for the EN plots and 1431 to 1596 µg C g$^{-1}$ soil at the CS sites. The minimum values were studied in the pyrogenic eluvial horizon ($E_{pyr}$): 8.5–83 for CS and 9–261 µg C g$^{-1}$ soil in the EN plots. At the same time, the QR values in this horizon, on the contrary, increased (QR = 2.7) (Figure 6), indicating the stress of the microbial communities. It was noted that, in general, the saturation of EN podzols with the microbial biomass is higher than in the CS plots. On average, the MB content in the CS pits was 42 µg C g$^{-1}$ soil, and in EN, 132 µg C g$^{-1}$ soil. The microbial carbon contribution to organic carbon ($C_{mic}$ at $C_{tot.}$) increased with depth. The $C_{mic}$ share at $C_{tot.}$ in CS podzols was 0.2 to 0.72%, and in EN, 1.3 to 3.0% (Figure 6).

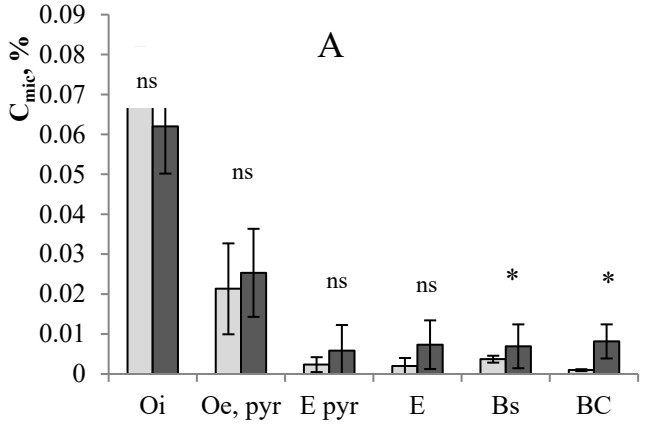
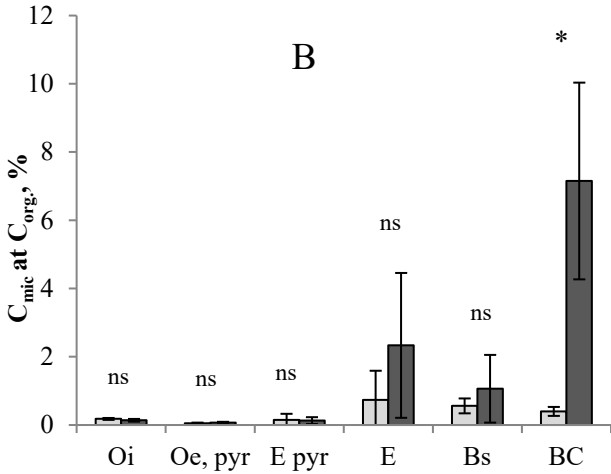
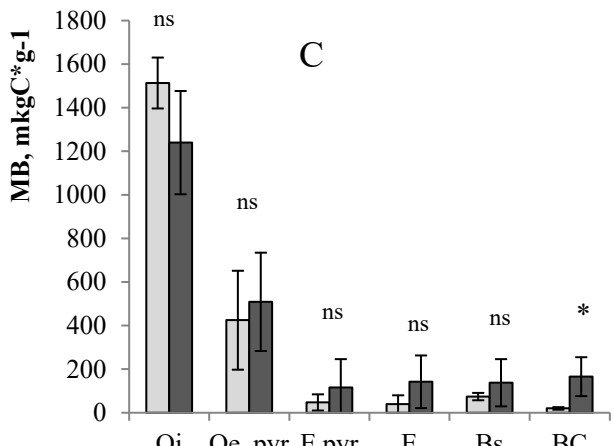
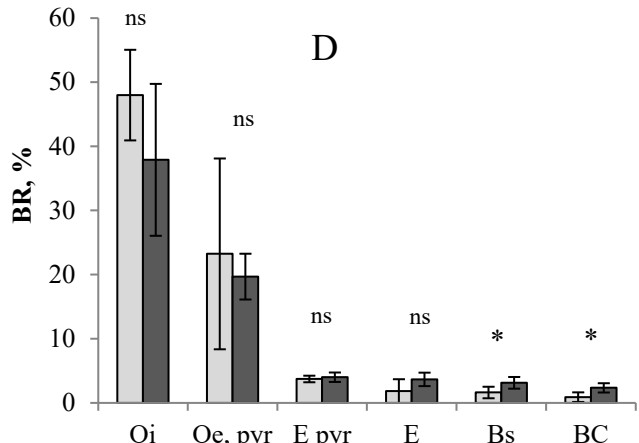
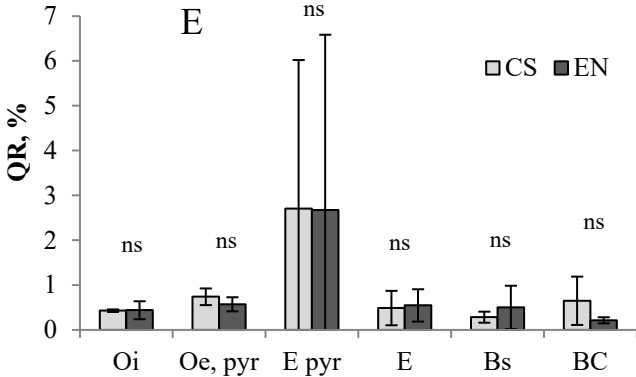

**Figure 6.** Values of microbial carbon, % (**A**), part of $C_{mic}$ at $C_{tot.}$, % (**B**), microbial biomass (MB) (**C**), and basal respiration (BR) (**D**), and microbial metabolic coefficient (QR) (**E**), ($n = 3$, the error bars are the standard errors). Abbreviation: CS—Central Siberia, EN—European North. *—indicate significant differences between EN and CS, ns—differences between EN and CS are not significant at $p < 0.05$.

The development of bacteriobiomes and mycobiomes in pyrogenic horizons is closely related to the time of past fires. Differences in the qualitative composition of the main groups of fungi and bacteria in the soils of the studied sites (CS and EN) were noted (Table 4).

**Table 4.** Relative content of prokaryotes and fungi at the type level in samples from the studied sites, averaged data by horizons, %.

| Taxonomic Position, Phylum | Fraction of OTU from Total Number of Obtained Sequences (%) | | | | | | | | | | | |
|---|---|---|---|---|---|---|---|---|---|---|---|---|
| | Zotino (Krasnoyarsk Region), Central Siberia (CS) | | | | | | Pechora-Ilychsky Nature Reserve (Komi Republic), European North (EN) | | | | | |
| | 45 CS | | 79 CS | | 121 CS | | 109 EN | | 113 EN | | 131 EN | |
| | Horizon, Depth, cm | | | | | | | | | | | |
| | Oe,pyr 3–5 | Epyr 5–11 | Oe,pyr 1–2 | Epyr 2–6 | Oe,pyr 0–3 | Epyr 4–10 | Oe,pyr 2–4 | Epyr 4–7 | Oe,pyr 2–4 | Epyr 4–7 | Oe,pyr 1–3 | Epyr 6–7 |
| Bacteria | | | | | | | | | | | | |
| *Proteobacteria* | 27.68 | 27.04 | 28.31 | 20.14 | 20.14 | 15.84 | 17.98 | 19.77 | 24.59 | 21.3 | 28.38 | 18.66 |
| *Actinobacteria* | 11.86 | 9.72 | 17.97 | 7.77 | 10.31 | 9.6 | 13.14 | 7.9 | 9.03 | 9.26 | 24.57 | 15.05 |
| *Acidobacteria* | 12.1 | 8.62 | 10.35 | 7.49 | 8.7 | 6.61 | 9.63 | 10.78 | 9.02 | 8.78 | 15.91 | 6.24 |
| *Planctomycetes* | 8.17 | 16.86 | 8.69 | 21.86 | 9.2 | 13.52 | 16.09 | 13.1 | 18.55 | 10.57 | 5.97 | 16.43 |
| *Verrucomicrobia* | 1.55 | 2.28 | 0.89 | 2.21 | 3.41 | 1.3 | 0.65 | 2.19 | 0.95 | 0.88 | 0.26 | 1.02 |
| *Bacterioidetes* | 2.42 | 1.49 | 2.52 | 0.35 | 0.84 | 1.05 | 0.36 | 0.86 | 1.07 | 1.59 | 0.67 | 0.66 |
| *Cyanobacteria* | 0.14 | 0.16 | 0.26 | 0.16 | 0.31 | 0.13 | 0 | 0.05 | 0.06 | 0.09 | 0.095 | 0.03 |
| *Firmicutes* | 0.09 | 0.31 | 0.37 | 1.01 | 0.35 | 0.61 | 0.02 | 0.1 | 0.81 | 0.28 | 1.31 | 1.73 |
| *Myxococcota* | 0.3 | 0.27 | 0.12 | 0.16 | 0.52 | 0.41 | 0.35 | 0.38 | 0.67 | 0.18 | 2.25 | 0.64 |
| Unc._Bacteria | 1.39 | 1.03 | 1.45 | 0.76 | 1.54 | 0.62 | 0.9 | 0.87 | 1.89 | 0.77 | 0.14 | 0.28 |
| Fungi | | | | | | | | | | | | |
| *Ascomycota* | 90 | 59 | 75 | 66 | 62 | 74 | 79 | 63 | 99 | 80 | 94 | 83 |
| *Basidiomycota* | 2 | 3 | 20 | 9 | 7 | 0.6 | 12 | 10 | 0.2 | 2.6 | 0.7 | 6 |
| *Mortierellomycota* | 0.07 | 0.02 | 0.3 | 0 | 0.08 | 0.04 | 1.18 | 11 | 0.3 | 1.7 | 0 | 0.09 |
| *Mucoromycota* | 6 | 36 | 2.3 | 25 | 28 | 26 | 6 | 11 | 0.5 | 14 | 5 | 9 |
| unc_Fungi | 1.5 | 1.3 | 2.1 | 0.09 | 3 | 0.05 | 1.2 | 4.5 | 0.06 | 1.1 | 0.11 | 0.5 |

Abbreviation: CS—Central Siberia, EN—European North. The number before the abbreviation means how many years have passed since the last fire.

Representatives of the *Ascomycota* phylum dominated among the fungi in all CS and EN sites. In the EN sites, the representatives of *Basidiomycota* were the sodominants (up to 12%), and *Mucoromycota* (up to 26%) in the CS sites, which were most actively developing in the $E_{pyr}$ horizon. *Proteobacteria, Actinobacteria, Acidobacteria,* and *Planctomycetes* phylum representatives dominated among the bacteria.

In the $O_{e,pyr}$ horizon of the CS plots, their share was 48%–65%, and in the $E_{pyr}$ horizon, this was 46%–62%. In the EN plots in the $O_{e,pyr}$ horizon, the share was 57%–75%, and in the $E_{pyr}$ horizon, it was 50%–56% (Table 4). It is noted that the contribution of $C_{mic}$ to $C_{tot.}$ in EN exceeded that in CS by 4.4 times. The indicator ($C_{mic}/C_{tot.}$) has a correlation coefficient of r = 0.52, $p < 0.05$ with SSA. Microbiological indicators of MB and BR are statistically significantly correlated with the exchange cations (r = 0.52–0.85, $p < 0.05$), cations exchange capacity (r = 0.70–0.73, $p < 0.05$), and content of WEOC (r = 0.79–0.85, $p < 0.05$) and WEON (r = 0.80–0.85, $p < 0.05$). In addition, there is a tendency of correlation with the total carbon content (r = 0.45–0.47, $p < 0.05$) and negative values of the correlation coefficient with the pH of both water and salt (r = $-0.43 \ldots -0.58$, $p < 0.05$).

The microbiological activity was significantly influenced by soil pH, $C_{tot.}$, and $N_{tot.}$ content. In the CS podzols, the values of MB and BR correlated with pH (r = $-0.64$ and r = 0.72), $C_{tot.}$ (r = 0.89 and 0.96), and N (r = 0.88 and r = 0.95, respectively). In the EN podzols, the values of MB and BR correlated with pH (r = $-0.66$ and r = $-0.74$), $C_{tot.}$ (r = 0.91 and 0.96), and $N_{tot.}$ (r = 0.92 and r = 0.95, respectively) as well.

Statistical analysis by principal components (PCA) based on the soil parameters of the first three soil layers of the pits showed a difference between them and a significant influence of the pyrogenic factor (Figure 7). The first component accounted for 59.9% of the total variance and the second component for 20.1%. The first component was determined by the content of carbon (C). The $C_{mic}/C_{org}$ ratio made the largest contribution to the second component. There are three clusters in the area of the first two components. The first cluster, located in the area of the second component, contained pyrogenic E layers of both the EN and CS pits. The second cluster, located in the area of the first component,

contained non-pyrogenic Oi layers of two soil types. The third cluster was formed by pyrogenic O$_{e,pyr}$ layers.

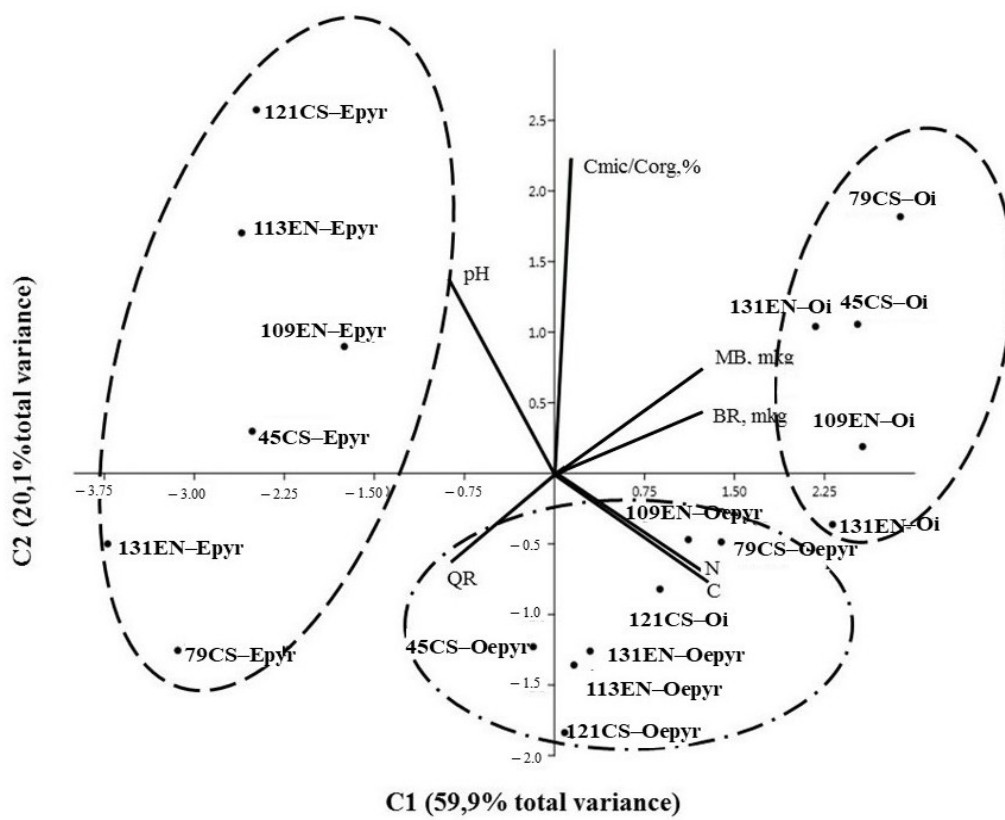

**Figure 7.** Position of different soil horizons in the plane of the two principal components obtained by the main chemical and microbiological characteristics (averaged data by horizons). Abbreviation: CS—Central Siberia, EN—European North.

## 4. Discussion

### 4.1. Vegetation Dynamics and Dendrochronology

The total projective cover (TPC) of lower vegetation layers depends on the age of the last fire. The plots located at Komi burned 109 or more years ago (Table 1), and the lower layers of the plant communities are currently characterized as well developed. The dwarf shrub-herb TPC ranged from 30 to 60% and the moss-lichen TPC from 80 to 90%. Compared to Komi, in the Krasnoyarsk region at plots that burned 45 and 79 years ago (45 CS and 79 CS), the lower vegetation layers are poorer, have lower TCP, and contain less species (Table 1). We assume that the age of the last fire is crucial for the lower vegetation layer's development. Plot 121 CS differs from the other plots by a very poor moss-lichen layer. This fact might be evidence of relatively low soil moisture, which in turn led to the low TPC of the dwarf shrub-herb layer (<5%) at this plot.

Throughout all study plots, TPC of the dwarf shrub-herb layer was found to be minimal at the plots that burned relatively recently and then increased during post-pyrogenic succession. In boreal forests, the middle-term post-fire recovery rate of the TPC [55] and biomass [56] of the moss-lichen layer is much faster than the other vegetation layers. In our study, we also revealed relatively fast restoration of the moss-lichen layer at all plots (Table 1) while a stable state was maintained during further succession.

In the studied forests of Komi Republic located at the foothills of the Ural Mountains, the average fire periodicity is 79 years. On the plains, the fire return interval is much higher, ranging from 12 to 106 years (average 55). Recent studies on the dynamics of fires in the territory of the Pechora-Ilych Reserve and its adjacent territories [57] revealed a 46-year fire cycle during the period from 1650s. Studies in northern Scandinavia [58,59] reported an

average fire cycle ranging from 40 to 300 years. The swampy pine forests from the northern taiga in the east part of the Urals have burnt up to 10 times over the past 100 years with a 3–30 year periodicity [60]. Gorshkov et al. [61] reported a 100 year average frequency of fires for the pine forests of the European North. According to Listov [62], the fire period was 20–50 years in the lichen pine forests of the Mezen River Basin.

The fire return intervals (FRIs) in the middle taiga zone of CS were estimated to be 25 to 50 years [63]. According to Kharuk et al. [64], FRI (in larch forests) tends to increase northward from $112 \pm 49$ years at 65 °N to 300 years at ~71 °N altitude. The landscape position is also very important, in that southern slopes burn more often and swampy valleys much less often [65].

Thus, it was found that FRIs between fires and the vegetation of the ground cover of the studied pine forests of the European North and Central Siberia are close to each other. The moss-lichen layers are almost completely restored 45 years after the fire.

### 4.2. Morphological and Chemical Properties

The studied soils are characterized by a typical morphological structure with the pyrogenic features in the upper genetic horizons, which is similar to the literature data on the podzols of the Leningrad region, Karelia, Central Yakutia, Finnish Lapland, and sandy soils of the Baikal region [66–69]. Under low-intensity surface fires, there is a change in the dominant vegetation cover and burning organic horizon. As a result, the pyrogenic $Q_{pyr}$ horizon is formed in the soils of the young burnt-out forest instead of the organic horizon [30,31]. Most of the pyrogenic traits in the soils of old-growth forests are concentrated in the lower part of the litter and the upper mineral horizon. Similar results are typical for the soils of pine forests in the Baikal region [70], which confirms the influence of surface fires on the transformation of organic soil horizons into new specific pyrogenic horizons.

Important visible parameters of the fires' impact are the thickness and humidity of the organic soil horizons. The minimum values of the moisture content and thickness of the organic horizons were found on 45 CS areas, on which the moss-lichen layer of plants had not yet completely formed. With the time that has passed since the fire, there is a clear trend of an increase in the thickness of the ground cover and the moisture content of the organic soil horizons. The exception is the section 79 years after the fire, in which the moisture content is low. This is due to the preservation of a pronounced dominance of lichens in the ground cover of a specific sample area due to selective and partial burnout. An important diagnostic role in the analysis of the duration of time since the last fire is played by the thickness of the organic horizon. Our data correspond to the results given in the literature [61]: the average thickness of forest litter during the stabilization stage is 7.5–8.5 cm for pine forests.

An important factor in the transformation of the organic horizon is the combustion temperature of plant materials [16]. According to Santín and Doerr [4], at temperatures of about 300 °C, a fire can change the structure of organic matter complexes, forming pyrogenic compounds (alkaline ash and charcoal). Charcoal particles play an important diagnostic role in genetic horizons [69,70]. It was revealed that pyrogenic morphological features (charcoal, soot, partially charred residues) can be traced not only in the organic but also in the upper mineral horizons of soils. The eluvial horizons can accumulate pyrogenic carbon. As a result of fire, the upper mineral horizons are enriched with pyrogenic residues ($E_{pyr}$). The presence of well-diagnosed coal particles in the litter and eluvial horizons was revealed in all the studied post-pyrogenic soils, which persists for a long time after a fire, even after 131 years.

The chemical properties of the studied EN and CS soils are very similar. Podzols are among the most acidic soils in the world. The maximum acidity is characteristic of the lower subhorizons of the litter and the eluvial horizon. The content of carbon and nitrogen in the $E_{pyr}$ horizon is significantly higher compared to the E horizon for all studied soils. This may be associated with the enrichment of the upper part of the horizon with pyrogenic

products, their migration, and accumulation. As vegetation recovers, the soil acidity values of the post-pyrogenic areas approach strongly and moderately acidic values.

SSA is a complex parameter that depends on the soil texture, cation exchange capacity, and other physical properties. SSA has significant correlation values with the content of exchange cations $Ca^{2+}$ (r = 0.96, $p < 0.05$), $Mg^{2+}$ (r = 0.97, $p < 0.05$), $K^+$ (r = 0.84, $p < 0.05$), $Na^+$ (r = 0.91, $p < 0.05$), their sum (r = 0.97, $p < 0.05$), and CEC (r = 0.74, $p < 0.05$). In addition, SSA is significantly correlated with the granulometric composition. The strongest SSA relationships were found with silty (r = 0.94–0.97, $p < 0.05$) and clay fractions (r = 0.93, $p < 0.05$). Negative significant correlation coefficients were established with the sand fractions (r = from $-0.58$ to $-0.65$, $p < 0.05$).

### 4.3. Soil Organic Matter of Studied Soils

Water-soluble organic compounds are represented by the most reactive substances. The quantitative and qualitative composition is largely determined by the plants of the ground cover and corresponds to the type of moisture and the type of forest [31,41]. This fact is probably due to the destruction of organic material concentrated in the forest organic horizons: the main source of water-soluble organic substances in the upper layers of the soil. The post-pyrogenic change in the carbon content of water-soluble compounds is associated with two main factors. The first is the production of water-soluble carbon by ground cover plants. Probably, this factor is crucial in identifying the patterns of pyrogenic effects and subsequent changes in post-pyrogenic successions. After a fire, a significant part of the plants dies. In the course of their further restoration, an increase in the contents of carbon and nitrogen of water-soluble organic compounds is observed due to the influx of organic substances from the restored organic horizons. The second important factor is the formation of a high proportion of pyrogenic inclusions (coals), which can absorb a significant part of the organic compounds entering the soil [71–73].

It is known that fires change the chemical composition of organic matter, leading to the enrichment of soils with pyrogenic black carbon (PyC) [74–77]. Comparing our results with the data of the podzols of Finnish Lapland [68], it can be assumed that part of the PyC as a result of oxidation passes into the composition of WSOC in soils [78,79]. However, this requires additional studies of the molecular composition of WSOC.

Forest fires are recognized as the main natural source of PAHs [80]. PAHs are formed during the combustion process. An increased content of light PAHs (about 70% of the total mass fraction of PAHs) in the organic horizon immediately after fire was also revealed by other authors [25]. According to Campos et al. [81], fire immediately caused an increase in the PAH (mainly light compounds) levels in the upper soil layer, but after four months, this effect largely disappeared. According to [80,81], PAHs can be released from the soil during the first year after a fire. Our earlier studies suggest that high levels of light PAHs may remain in the litter 10–16 years after a fire [22]. The decrease in the proportion of light PAHs in soils with older fires is due to the decomposition of naphthalene. Some authors claim that light PAHs (naphthalene, anthracene, chrysene, fluorene, and fluoranthene) tend to move from the soil to the air [82]. The increased contribution of naphthalene in the organic soil horizon of the 121-year-old site (121 CS) after the fire is due to the overall low PAH content. Perhaps, in this case, the processes of decomposition of three to four nuclear structures of PAHs to lighter ones occurred since the proportion of high-molecular PAHs at this site had maximum values. The high content of heavy polyarenes in the soils of this site (121 years) could be due to a significant number of fires. Since 1663, five fires have occurred, which contributed to the active accumulation of heavy structures, mainly benz [b]fluoranthene and dibenz [a, h]anthracene, in the organic horizon. Significant concentrations of benz [b]fluoranthene were found in pyrogenic soil of the Komi Republic 131 years after the fire (131 EN), for which an increased frequency of fires was also noted. At the same time, dibenz[a, h]anthracene was absent in the organic horizon, which could be due to the old age of the site. Because of this, dibenz[a, h]anthracene could already decompose to lighter structures. The same works have shown the ability of the soil

microbiota to use low-molecular-weight PAHs as an energy source and, in their absence, heavier structures [81–83]. In addition, an increased content of indenopyrene in the soils of the Komi Republic was found in podzols for which fires occurred one to two times, which could be associated with the processes of pedogenesis more than with fires.

Studies have shown that the combustion conditions and properties of the organic horizon strongly affect the PAH levels [25,81]. They suggest that the formation of low-molecular-weight PAHs depends more on the fire conditions. The content of heavy PAHs is mainly related to the characteristics of the biomass, species diversity, and the type of organic horizon. Thus, the organic horizon formed by *Pinus nigra* was characterized by the lowest PAH content. Whereas the combustion of *Pinus pinaster* revealed large amounts of PAHs in the organic horizon [25]. We calculated a number of coefficients that are recommended for identifying the contribution of pyrogenic and pedogenic PAHs [84–91]. However, we did not find significant correlations with the time of the last fire. On the one hand, this may be due to the fact that the diagnostic ratios of PAHs were originally developed for the snow cover [84,92]. Therefore, they may not work for soils due to the presence of many interfering influences. On the other hand, the pyrogenic factor could be based on the processes of anthropogenic burning of oil and gas. So, natural biomass combustion PAHs might be included in petrogenic PAHs.

The low content of PAHs in the mineral horizons of soils is largely due to the weak ability of PAHs to migrate and their low solubility [93]. Heavy PAHs, characterized by lower solubility [94], were absent or were present in minimal concentrations in the mineral horizons. Similar data were obtained for different soil types [95], including for soils under medium-taiga shrub-green-moss pine forests in the first months after fire [11]. Under these conditions, PAHs were mainly concentrated in the soil organic horizon. Other authors have shown the possibility of PAHs leaching into the underlying horizons for the example of soils in the zone of action of a coke factory [96]. The detection of heavy PAHs in the mineral horizons of soils of the Komi Republic could be associated with climatic factors and possibly a more intensive washing regime (more precipitation, snowmelt) than in the Krasnoyarsk region. It can be assumed that the higher intensity of fires, and hence the combustion temperature characteristic of Siberia, leads to a lower content of PAHs.

### 4.4. Microbiological Properties

Thus, the podzols of the CS and EN pine forests differed in terms of the MB content, specific respiration rate, and mineralization processes. In the soil of the EN pine forests, the MB content in the mineral horizons was, on average, three times higher than in CS. In all pine forests, the maximum MB and BR values were recorded in the litter, where not only microorganisms but also other biota (mosses, lichens, microfauna) take part in the respiration processes. In the $E_{pyr}$ horizon, the minimum values of MB and rather high respiration (BR, QR) were noted, which indicates stressful conditions for microbocenosis. Most likely, the soil in these horizons has not recovered after the fire, there is little organic matter ($C_{tot.}$ and $N_{tot.}$) in it, and the activity of microbial communities is reduced. In the CS podzols, along with *Ascomycota*, *Mucomycota* fungi most actively colonized the $E_{pyr}$ horizon, and in EN, *Basidiomycota.* Among the bacteria, representatives of *Planctomycetes*, *Verrucomicrobia*, and *Firmicutes* significantly contributed to the development of $E_{pyr}$.

PCA showed that the C, N, and MB contents contributed most to the differences between the EN and CS soil types, and the differences between the pyrogenic and non-pyrogenic layers were due to the $C_{mic}/C_{org}$ (%) ratio and pH values.

### 5. Conclusions

The results of a comparison of the basic parameters of natural soils, which are not affected by any types of human activity (except fires), of forest pine ecosystems of the EN and CS regions, which can be used as background for comparison, were presented. This study revealed similar morphological and physicochemical characteristics of the albic

podzols in the EN and CS regions. It was shown that pyrogenic signs have persisted in the organic and upper mineral soil horizons for more than 130 years after the last fire.

The soils of the studied regions did not statistically differ in their physicochemical properties and profile distribution of indicators (pH, BS, CEC, texture, carbon and nitrogen contents and storages, content of water-soluble forms of carbon and nitrogen). In the upper mineral horizon $E_{pyr}$ of the European North, a statistically greater value of the total PAH content was revealed.

The microbiological activity of the organic horizons of the EN and CS soils was similar. Podzols were characterized by high MB and BR values in the organic horizons and a low MB content in the upper mineral soil horizons. The mineral horizons of the EN and CS soils differ in the microbial biomass content, specific respiration rate, and mineralization processes. The MB content in EN was, on average, three times higher than in CS. The lower EN mineral horizons were also characterized by higher values of microbial carbon and its contribution to total carbon. Differences were also noted in the qualitative composition of the studied soils' microbiomes in EN, which are represented by a wide variety of bacteria and fungi. This is probably due to the more continental conditions for the formation of podzols in the CS territory compared to EN.

**Supplementary Materials:** The following supporting information can be downloaded at: https://www.mdpi.com/article/10.3390/f13111831/s1, Figure S1: Profile photos of the investigated podzols; Table S1: texture of studied soils.

**Author Contributions:** Conceptualization, A.A.D. and A.S.P.; methodology, A.A.D., I.D.G. and A.S.P.; software, V.V.S.; validation, V.V.S., A.A.D. and I.D.G.; formal analysis, A.A.D., I.D.G., E.V.Y., V.V.S., I.N.K., E.Y.M. and Y.A.D.; investigation, A.A.D., I.D.G. and V.V.S.; resources, A.A.D., A.S.P. and I.D.G.; writing—original draft preparation, A.A.D., V.V.S., E.V.Y., Y.A.D. and I.D.G.; writing—review and editing, A.A.D., I.D.G., V.V.S. and A.S.P.; visualization, V.V.S., I.D.G. and A.A.D.; supervision, A.A.D.; project administration, A.A.D.; funding acquisition, A.A.D. All authors have read and agreed to the published version of the manuscript.

**Funding:** This work was supported by the Russian Foundation for Basic Research (RFBR) under Grant No. 19-29-05111 mk and budgetary theme of IB FRC Komi SC UB RAS 122040600023-8.

**Institutional Review Board Statement:** Not applicable.

**Informed Consent Statement:** Not applicable.

**Data Availability Statement:** The data presented in this study are available upon request from the corresponding author.

**Conflicts of Interest:** The authors declare no conflict of interest.

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
