# Peer review of "Albic Podzols of Boreal Pine Forests of Russia: Soil Organic Matter, Physicochemical and Microbiological Properties across Pyrogenic History"

_forests, doi:10.3390/f13111831_

Round 1

Reviewer 1 Report

This is an interesting story related to the effects of fire on soil chemical and microbial properties. After reviewing, some parts were needed been improved especially in material and methods. The results were mainly described with on any statistical analysis. Some details about microbial community were added, such as the PCR conditions for different primer sets and the sequencing analysis. The results had no main idea which connected the whole story. The discussion included too many results of your own study.  

Specific comments:

Line 87:20*20 m2?

Line 175-176:the PCR conditions were different for different primer sets (bacteria, fungi and archaea).

Line 188: different microbial community had different reference database. In general, The Unite database was used to analyze the fungal community.

The results lack statistical analysis, such as Fig. 2, 3, 4 and 5.

The microbial structure may be added using NMDS or PCoa.

Line 552-564: This part was result rather than discussions. Similarly, line 52-537.

The correlation analysis can’t achieve the effects of soil properties and fire on microbial community. I suggest you can try to use the SEM to connect the factors on soil microbial community diversity and structure.

Author Response

We thank the reviewer for valuable advice and comments on our article. We have tried all comments to take into account and correct. We believe that your comments have benefited this article.

Point 1: Line 87:20*20 m2?

Response 1: Text has been edited

Point 2:

the PCR conditions were different for different primer sets (bacteria, fungi and archaea).

Line 188: different microbial community had different reference database. In general, The Unite database was used to analyze the fungal community.

Response 2: Text has been edited

Point 3: The results lack statistical analysis, such as Fig. 2, 3, 4 and 5. The microbial structure may be added using NMDS or PCoa.

Response 3: Yes, thanks, we have added another figure with PCoa.

Point 4: Line 552-564: This part was result rather than discussions. Similarly, line 52-537.

Response 4: Text has been edited.

Point 5: The correlation analysis can’t achieve the effects of soil properties and fire on microbial community. I suggest you can try to use the SEM to connect the factors on soil microbial community diversity and structure.

Response 5: Thank you. This will increase the volume of the article too much. We will test these approaches in the future.

Thank you!

Alexey A. Dymov, Irina D. Grodnitskaya, Evgenia V. Yakovleva, Yuri A. Dybrovskiy, Ivan N. Kutyavin, Viktor V. Startsev, Evgeni Yu. Milanovsky and Anatoly S. Prokushkin

Reviewer 2 Report

The manuscript "Albic Podzols of boreal Pine forests of Russia (Komi and Krasnoyarsk regions): physico-chemical, microbiological properties and soil organic matter across pyrogenic history" is very well written and suitable for publication in Forests. However, some corrections are needed.

Abstract: The methodology is not clear and should help the reader.

Material and Methods

Why these soil names 109EN, 113EN, 131EN1 and 45CS, 79CS, 121CS? It was not clear to the reader. What are the coordinates of the site?

Figure 1: The images on the left represent what? Are they enlargements of a point? It has to be self-explanatory.

Line 121: "according to [40]." Who?

Data statistician: How were the data analyzed and presented? You only report the correlation.

Results

Lines 198-203: All scientific names should be in italics.

Tables 1 and 2. "Site": This nomenclature was not clear to me.

Lines 237 and 238: Use pH(H2O) with the number 2 superscripted.

Line 238: Figure 3A or Figure 2A? It is confusing, correct.

Figure 2. You should write the name of the y-axis of both figures, not just the unit of measurement.

Figures 3, 4, 5, 6. You must write the name of all x- and y-axes and their abbreviations.

Table 3. Did you only do one analysis? No error or standard deviation of the mean?

Conclusion

It is very well written and answers the paper's objective.

Appendices should not be here in the paper.

References

All of them are adequate, although some are in Russian and I couldn't read them.

Author Response

We thank the reviewer for valuable advice and comments on our article. We have tried all comments to take into account and correct. We believe that your comments have benefited this article.

Point 1: Abstract: The methodology is not clear and should help the reader.

Response 1: The text has been edited.

Point 2: Material and Methods

Why these soil names 109EN, 113EN, 131EN1 and 45CS, 79CS, 121CS? It was not clear to the reader. What are the coordinates of the site?

Response 2: We have added explanations to the text.

Point 3: Figure 1: The images on the left represent what? Are they enlargements of a point? It has to be self-explanatory.

Response 3. The Figure has been edited.

Point 4: Line 121: "according to [40]." Who?

Response 4: Paraphrased and edited the text

Point 5: Data statistician: How were the data analyzed and presented? You only report the correlation.

Response 5: In this manuscript, we analyzed means and standard deviations. They are shown in the figures.

Point 6: Results

Lines 198-203: All scientific names should be in italics.

Response 6: Corrections have been made to the text.

Point 6: Tables 1 and 2. "Site": This nomenclature was not clear to me.

Response 6: We have edited the text and abbreviations.

Point 7: Lines 237 and 238: Use pH(H2O) with the number 2 superscripted.

Point 7: We have edited the text.

Point 8: Line 238: Figure 3A or Figure 2A? It is confusing, correct.

Point 8: In this case, the link is correct, Figure 3. We have changed the location of the figures.

Point 9: Figure 2. You should write the name of the y-axis of both figures, not just the unit of measurement.

Point 9: We have added the name of the axes.

Point 10: Figures 3, 4, 5, 6. You must write the name of all x- and y-axes and their abbreviations.

Point 10: We have added the name of the axes.

Point 11: Table 3. Did you only do one analysis? No error or standard deviation of the mean?

Point 11: The samples were analyzed once. However, in view of satisfactory precision and accuracy results (precision and accuracy analysis was carried out for each 10 samples), we consider that the measurements are correct and the results correspond to the values of PAH content in the studied soil horizons. Since mixed samples of the respective soil horizons were used for the analysis, it can be stated that the results reflect the average characteristics of a particular soil layer. As required by the reviewer, relative errors are given in Table 3 according to the measurement methodology.

Point 12: Appendices should not be here in the paper.

Point 12: We have made a file with Supplementary.

Thank you!

Alexey A. Dymov, Irina D. Grodnitskaya, Evgenia V. Yakovleva, Yuri A. Dybrovskiy, Ivan N. Kutyavin, Viktor V. Startsev, Evgeni Yu. Milanovsky and Anatoly S. Prokushkin

Round 2

Reviewer 1 Report

The results lack statistical analysis. The authors didn't correct it. 

Author Response

Dear reviewer!

We have added information on statistical significance between soil properties in the study regions (EN vs CS) on Figures 2-6.